# TRAINING INSTABILITY AND DISHARMONY BETWEEN RELU AND BATCH NORMALIZATION

## ABSTRACT

Deep neural networks based on batch normalization and ReLU-like activation functions experience instability during early stages of training owing to the high gradient induced by temporal gradient explosion. ReLU reduces the variance by more than the expected amount and batch normalization amplifies the gradient during its recovery. In this paper, we explain the explosion of a gradient mathematically while the forward propagation remains stable, and also the alleviation of the problem during training. Based on this, we propose a Layer-wise Asymmetric Learning rate Clipping (LALC) algorithm, which outperforms existing learning rate scaling methods in large batch training and can also be used to replace WarmUp in small batch training.

## 1   INTRODUCTION

The success of deep neural networks is based on their great expressive power, which increases exponentially with depth (Chatziafratis et al., 2019). However, the deep hierarchical architecture of a deep neural network may induce the exploding/vanishing gradient problem (Pascanu et al., 2013), which degrades performance and may even make training impossible.

Several techniques have been suggested to address the aforementioned problem, including better weight initialization (Glorot & Bengio, 2010; He et al., 2015), activation functions (Nair & Hinton, 2010; Klambauer et al., 2017), residual connections (He et al., 2016), normalization methods (Ioffe & Szegedy, 2015; Ulyanov et al., 2016; Nam & Kim, 2018; Ba et al., 2016; Wu & He, 2018; Qiao et al., 2019), etc. However, Contrary to popular belief, gradient explosion is present in various modern deep learning architectures that use batch normalization and ReLU-like activation functions (Figure 1). The stable flow of forwarding activation does not guarantee the stable flow of backward gradient when an entropy difference exists between the layers (Philipp et al., 2017).

A gradient explosion in modern deep learning architecture has been reported in a number of papers (Philipp et al., 2017; Frankle et al., 2020; You et al., 2017), but its cause and solution have not been sufficiently researched. (Philipp et al., 2017) discussed the context and presented an intuitive understanding of the problem, considering the modality of gradient explosion while activation remains stable, the increase in entropy induced by the activation function, the alleviation of the problem using residual learning, etc. However, although the authors observed that gradient explosion occurs only in the case of batch normalization, they did not explain this phenomenon. They speculated that the sampling error of the normalization procedure amplifies the problem—however, no clear correlation has been reported between the gradient explosion rate and the expected sampling error (batch size, difference normalization scheme, etc.). To the best of our knowledge, this is the first attempt to demonstrate how disharmony between activation function and batch normalization causes gradient explosion and training instability during the early stages of neural network training mathematically. The alleviation of the problem during training is also discussed.

Exploding/vanishing gradient is a well-known problem in deep learning. It may seem unnatural that it still exists but is not widely known. One reason is that the problem is not as severe as before. The exploding rate is approximately $\sqrt{\pi/(\pi - 1)} \sim 1.21$ per effective depth. It is tolerable in networks with tens of layers or those with hundreds of layers and dense residual connections. Moreover, the exploding gradient at the initialization state is rapidly relieved during training, and even a vanishing gradient state can be approached. Thus, this problem has been referred to by different names in

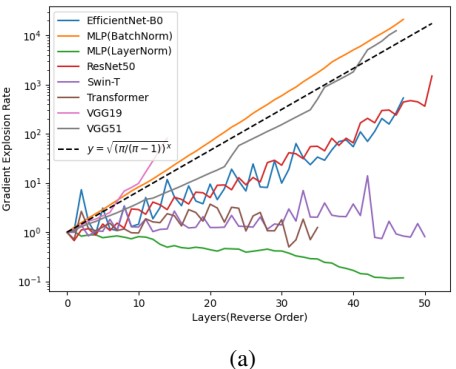 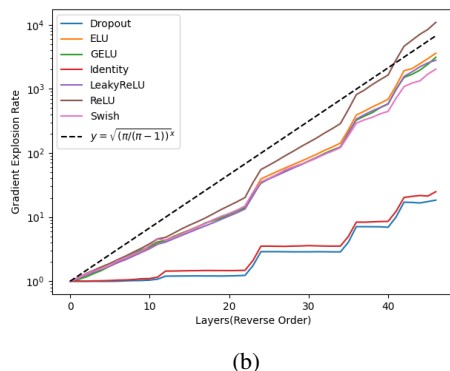

(a)                                   (b)

Figure 1: Gradient explosion rate ($\sqrt{Var(g^n)/Var(g^N)}$) of deep neural network models at the initialization state corresponding to (a) different architectures and (b) activation functions. The explosion rate is approximately $\sqrt{\pi/(\pi-1)}$ in vanilla networks with batch normalization; but it is lower in architectures with residual connections (He et al., 2016), which reduces the effective depth. Moreover, gradient explosion does not occur in architectures with layer normalization (Ba et al., 2016), including transformer-based architectures (Dosovitskiy et al., 2020; Liu et al., 2021). Figure (b) is plotted using a 51-layer VGG-like (Simonyan & Zisserman, 2014) architecture. Smoother variants of ReLU (Maas et al., 2013; Clevert et al., 2015; Ramachandran et al., 2017; Hendrycks & Gimpel, 2016) exhibit lower exploding rates as they exhibit flatter behavior near zero and lower signal blockage at the initialization state. In the extreme case, gradient explosion does not occur if the activation function is not used. Note that it also does not occur with DropOut (Srivastava et al., 2014), which can be regarded as a ReLU that blocks signals randomly.

the literature, such as 'large gradient magnitude at the early stage of training' or 'instability of deep neural network at the early phase of training' (Frankle et al., 2020; Goyal et al., 2017b). It is also often treated as an 'instability problem of large batch training' (Goyal et al., 2017b) since the problem becomes severe when a large batch size (and high learning rate, accordingly) is used during training. Although training instability occurs only during early stages of training, it can disrupt the balance between the layers and degrade the final performance (Goyal et al., 2017b; You et al., 2017).

## 2   How Gradient Explosion Occurs

As weights are repeatedly multiplied during forward/backward propagation, the exploding or vanishing gradient problem is commonly believed to be caused by excessively large or small parameters (Bengio et al., 1994; Pascanu et al., 2013). Thus, it has been largely treated as the problem of initialization and maintenance of optimal weight scales. In that case, the problem would have been 'solved' with the advent of (batch) normalization (Ioffe & Szegedy, 2015), which automatically corrects suboptimal choices of weight scales. However, maintaining the norm of weights, and thereby the norm of forwarding propagation, does not automatically maintain the norm of backward propagation.

Rectified Linear Unit (ReLU) (Nair & Hinton, 2010) (as well as its smoother variants (Maas et al., 2013; Klambauer et al., 2017; Clevert et al., 2015; Ramachandran et al., 2017; Hendrycks & Gimpel, 2016)) blocks approximately half of the activation at each instance. In this sense, (He et al., 2015) assumed that it halves the output variance based on some zero-centered assumptions. Therefore, the authors concluded that initializing weights with $N(0, \sqrt{2/n_{out}})$ maintains similar variances in both forward and backward propagation.

However, the relationship of the activation function with the input should also be considered. As depicted in Figure 2, using the positive part of the activation is different from blocking randomly selected activations. In short, (He et al., 2015) described the situation assuming DropOut (Srivastava et al., 2014) to be the activation function instead if ReLU. In that case, both forward and backward signals are roughly halved, and neither exploding nor vanishing gradient occur. This is verified in

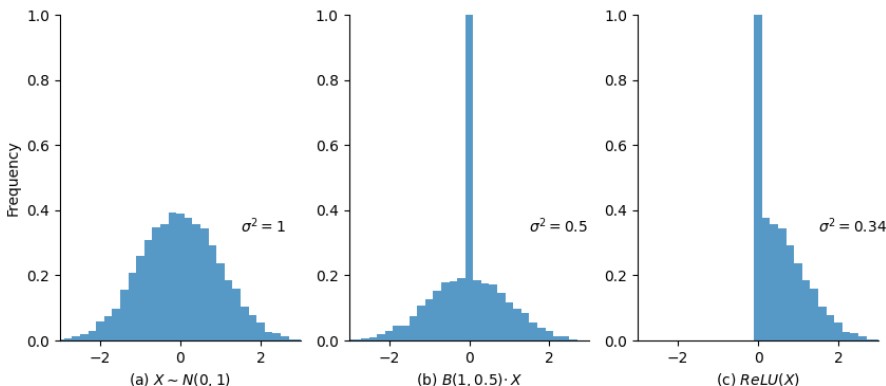

Figure 2: (a) Sample of normally distributed input. (b) Hypothetical output distribution assuming that ReLU randomly drops half of the input. (c) Real distribution after ReLU activation, which exhibits smaller variance than that in (b). (He et al., 2015) assumed the post-ReLU distribution to be similar to that depicted in (b), in which half the variance is compared to the input if the input is zero-centered. Since ReLU also halves the gradient during backpropagation, the authors concluded that selecting optimal weights to maintain input variance also guarantees stability of the variance of the gradient. However, the correlation between the input and the activation function may affect input variance and gradient variance differently. Thus, in batch normalization, which recovers the variance decreased by the activation function, the backpropagating gradient may increase exponentially while the forward propagation remains stable.

Figure 1(b). However, since the real activation functions are highly dependent on the input, it may affect the variance differently during forward and backward propagation.

**Proposition 1.** *Let $X = ReLU(Y) = max(Y, 0)$, where $Y \sim N(\mu, \sigma^2)$. Then, we have:*

$$E(X) = \mu(1 - \phi(-\frac{\mu}{\sigma})) + \frac{\sigma}{\sqrt{2\pi}}e^{-\frac{\mu^2}{2\sigma^2}} \tag{1}$$

$$Var(X) = (\sigma^2 + \mu^2)(1 - \phi(-\frac{\mu}{\sigma})) + \mu\frac{\sigma}{\sqrt{2\pi}}e^{-\frac{\mu^2}{2\sigma^2}} - (\mu(1 - \phi(-\frac{\mu}{\sigma})) + \frac{\sigma}{\sqrt{2\pi}}e^{-\frac{\mu^2}{2\sigma^2}})^2 \tag{2}$$

*where $\phi(x) = \int_{-\infty}^{x} \frac{1}{\sqrt{2\pi}}e^{-x^2/2}$ is the cumulative distribution function of the standard normal distribution.*

*Proof.* See Appendix A.1 □

**Theorem 1.** *Consider the repetitive neural network architecture, $f^n : \mathbb{R}^{d_n} \longmapsto \mathbb{R}^{d_{n+1}}$, where $x^{n+1} = f^n(x^n) = BatchNorm(W^n(ReLU(x^n)) + b^n) = \frac{(W^n(ReLU(x^n)) + b^n) - \mu}{\sigma}\gamma^{n+1} + \beta^{n+1}$, where $\gamma^{n+1}, \beta^{n+1} \in \mathbb{R}^{d_{n+1}}$ are affine transformation parameters and $\mu, \sigma \in \mathbb{R}^{d_{n+1}}$ are the estimated mean and standard deviation. Assume that inputs $(x_i^n)$ are independent and follow the zero-centered normal distribution and that the weights $(W_{ij}^n)$ are sampled from the zero-centered distribution. Moreover, let the gradient of the output $(g_j^{n+1} := dL/dx_j^{n+1}$, where $L$ denotes the loss) be not correlated to the input. Ignoring the sampling error of batch normalization, we have:*

$$E[\sum_i Var(x_i^n)Var(g_i^n) / \sum_j Var(x_j^{n+1})Var(g_j^{n+1})] \geq \frac{\pi}{\pi - 1} \tag{3}$$

*where the equality holds when the activation before the normalization layer exhibits identical variance.*

*Proof.* See Appendix A.2 □

If the size of activation remains similar, the gradient can be expected to increase exponentially at a rate of $\sqrt{\frac{\pi}{\pi-1}} \sim 1.21$

**Corollary 1.** *In addition to the assumption of Theorem 1, let $x^{n+1}$ and $g^{n+1}$ be zero-centered. Then, we have:*

$$E[\sum_i (x_i^n)^2 (g_i^n)^2 / \sum_j (x_j^{n+1})^2 (g_j^{n+1})^2] \geq \frac{\pi}{\pi-1} \tag{4}$$

*where the equality holds when the activation before the normalization layer exhibits identical variance.*

*Proof.* Trivial, since $Var(X) = E(X^2)$ for zero-centered distributions. □

The theorem can be roughly explained by the fact that the ReLU activation function reduces the variance by approximately $(\pi-1)/2\pi$ instead of $1/2$ under the zero-centered assumption. Without loss of generality, let the weight be initialized such that it maintains the variance after the transformation layer. Then, the sample variance before the normalization layer becomes $(\pi-1)/2\pi$ and the forward activation is divided by this value. During backpropagation, the gradient is multiplied by $2\pi/(\pi-1)$ at the normalization layer, and half are dropped at the activation function. Therefore the gradient is amplified at a rate of $\sim \pi/(\pi-1)$. The formula is much more complex without the zero-centered assumption, but the structure remains essentially identical. Since every activation function decreases variance (or increases entropy) by a greater amount than it blocks the gradient, it always amplifies the gradient *if the assumptions hold.*

### 2.1 Then Why does Gradient Explosion exist only during Early Stages of Training?

By Theorem 1, gradient explosion in a deep neural network seems to be unavoidable. However, in reality, gradient explosion exists only during early stages of training. The dynamics of a deep neural network change rapidly during the early stages of training.

Various factors contribute to gradient explosion or vanishing. For example, gradient explosion assigns a high gradient to low (near-input) layers and induces high variance in the distribution of weights of low layers, resulting in a mismatch of weight norms between layers. This contributes to the direction of the vanishing gradient. (Imagine that $y = w_2 w_1 x$, where $w_1 = 100$ and $w_2 = 1$.) On the other hand, gradient explosion is essentially induced by the interaction between batch normalization and activation function. Thus, it does not occur when the activation functions are linear or 'pseudo-linear' (Philipp et al., 2017). ReLU-like activation functions can be (partially) disabled if a node only outputs values with identical signs in most cases or $|\mu/\sigma| >> 1$. In the extreme case, the invalidated activation functions can be replaced with scalar values (0 or 1). Such a layer does not generate gradient explosion, although it does not function as a valid layer contributing to the effective depth either.

**Corollary 2.** *Under the assumptions of Theorem 1, let us consider inputs that are no longer zero-centered and $x_i^n \sim N(\mu_i^n, \sigma_i^n)$. At the limit where ReLU becomes completely pseudo-linear, we have:*

$$\lim_{\sigma_i^n/\mu_i^n \to 0 \, \forall i} E[\sum_i Var(x_i^n) Var(g_i^n) / \sum_j Var(x_j^{n+1}) Var(g_j^{n+1})] \geq 1 \tag{5}$$

*where the equality holds when the activation before the normalization layer exhibits identical variance.*

*Proof.* See Appendix A.3 □

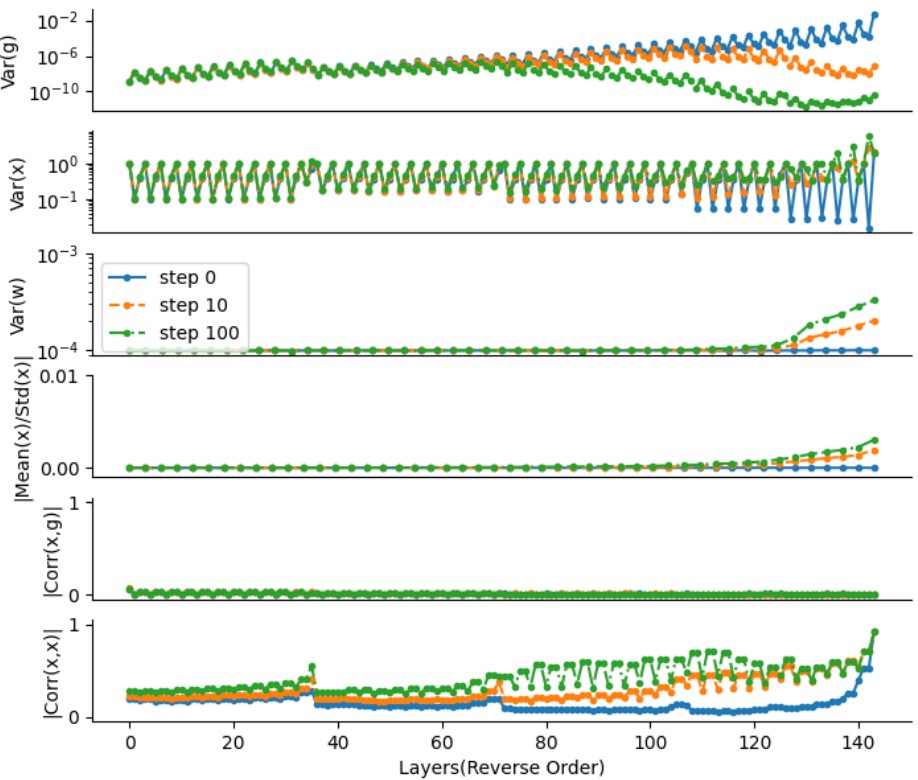

Figure 3: Sample chart of the variance of gradient/activation/weight, mean-std ratio, correlation between activation and gradient ($\sum_{i=1}^{C} |Corr(x_i, g_i)|/C$), and the average correlation between activations ($\sum_{i \neq j=1}^{C} |Corr(x_i, x_j)|/(C(C-1))$) during early stages of training. The order is reversed—the left side represents the layer near the output. All weights are initialized from $N(0, 0.01^2)$ for better visualization. Var(x) is normalized near 1 at every batch normalization layer. However, the variance is observed to increase after the ReLU and convolution layers during training, even when the norm of weights remains almost constant. Gradient explosion at the initialization state may be reduced by various factors, including an increase in the norm of weights near the input, pseudo-linearization of the activation functions, the correlation between the input and the gradient, etc. However, the correlation between activations seems to contribute to the decrease of the gradient primarily. It alters the variance after linear transformation and decreases the norm of the gradient flow while the forward flow remains stable since the variance of the sum of dependent variables is different from that of independent variables (See Theorem 2.)

However, the most fundamental factor that decreases gradient seems to be the correlation between input activations (Figure 3). Most of the assumptions in Theorem 1 seem to remain valid even after the neural network is trained. Internal variables such as weight, activation, and gradient, tend to remain zero-centered. And the correlation between activation and gradient also tends to remain small. However, a neural network tends to generate multiple redundant and duplicated signals during training (Anwar et al., 2017), which necessitates 'neural network pruning'. Since the variance of the sum of correlated random variables is different from that of independent random variables, even gradient vanishing may occur if a high correlation exists between the inputs.

**Theorem 2.** *Under the assumptions of Theorem 1, let us consider the case in which inputs are perfectly correlated. Then, we have:*

$$E[\sum_i Var(x_i^n)Var(g_i^n)/\sum_j Var(x_j^{n+1})Var(g_j^{n+1})] \geq \frac{\pi}{2+\pi} \tag{6}$$

*where the equality holds when the activation before the normalization layer exhibits identical variance.*

*Proof.* See Appendix A.4 □

## 3 SOLUTIONS

A large amount of the gradient induced by a gradient exploding is not an error of calculation. A change in the lower (near-input) layer actually creates a significant impact on the upper layers. Activation functions cannot be designed to avoid decreasing entropy as it compromises their function. Residual connections (He et al., 2016) decrease the effective depth and ameliorate the problem significantly, but not completely.

Fortunately, the problem is temporal. Although vanishing gradient may occur in the middle stages of training, it is less harmful than gradient explosion and practical observations suggest that no measures need to be taken to resolve it at this stage. In this sense, WarmUp (Goyal et al., 2017b) is an efficient and effective method to bypass the problem. It assigns a very small learning rate during the early stages of training, preventing the assignment of excessive noise to the neural network until the learning is stabilized.

But in more extreme cases, such as large batch training (and large learning rate, accordingly), a better solution is required. Layer-wise Adaptive Rate Scaling (LARS, Algorithm 1) (You et al., 2017) maintains the ratio between gradient and weight as a constant for each layer. It is a powerful and convenient tool for large-batch training and has been successfully applied in several large-scale experiments (Mathuriya et al., 2018; Narayanan et al., 2019; Chen et al., 2020a;b; Yu et al., 2022). Certain modifications have also been suggested. (You et al., 2019) modified the formula and applied the technique to the Adam (Kingma & Ba, 2014) optimizer to train a large-scale language model (Algorithm 2). (Huo et al., 2021) suggested using the average of the norm of the per-sample gradient instead of the minibatch gradient.

However, LARS and similar techniques have hardly been applied in small-batch experiments, since they restrict training and induce minor performance degradation. Even for large-batch training, (Nado et al., 2021) showed that we may obtain better results without using LARS if we can control the instability using WarmUp and careful hyperparameter tuning.

A large amount of the gradient means that a small change in that parameter affects the output (and near-output layers) significantly. But the gradient descent algorithm even assigns larger step sizes to that parameter, assuming the parameters are roughly connected in parallel and (joint) second-order differentials are negligible. However, since the neural network is highly hierarchical, this can make the training process noisy and unstable. The ideal solution would be a higher-order optimization algorithm, e.g., Newton's method (Newton, 1711), but it is not practical to obtain a Hessian matrix for a deep neural network with millions or billions of parameters.

Therefore, an engineering solution capable of assigning a smaller learning rate corresponding to instances of gradient explosion is required. Existing methods may suffice for this purpose, but they

are not very specific—WarmUp functions only during the early stages of training, but it reduces the learning rate for every layer with a fixed schedule. LARS reduces the learning rate when gradient explosion occurs, but it affects every layer during the whole training procedure. Moreover, it amplifies the learning rate when the gradient is excessively small, which may not be necessary.

We don't want a single step of gradient update causes more than a certain level of change in the output (and near-output layers). The amount of change would be proportional to the gradient norm times the learning rate, therefore the learning rate should be bounded by a value that is inversely proportional to the norm of the gradient. We found that the local learning rate of LARS (You et al., 2017) with a higher coefficient can serve as a suitable upper bound, better than the simple inverse of gradient norm.

Therefore we propose Layer-wise Asymmetric Learning rate Clipping (LALC), which sets an upper bound on the learning rate instead of scaling it. Algorithm 3 describes the algorithm, and Appendix B presents the PyTorch (Paszke et al., 2019) implementation. The changes are simple but it has desired properties—LALC does not amplify the gradient even when the gradient is significantly lower than the weight. Moreover, LALC is naturally disabled as learning stabilizes and a global learning rate decreases since it uses a fixed upper bound using a ratio between the weight and the gradient. The norm of the gradient after applying optimization techniques (e.g., momentum, weight decay, etc.) is calculated—the obtained value is slightly better than that obtained using the norm of the original gradient in our experiment.

## 4 EXPERIMENTS

The ResNet50 (He et al., 2016) model is used in both CIFAR10 (Krizhevsky & Hinton, 2009) and ImageNet (Deng et al., 2009) experiments. For LARS (You et al., 2017) and CLARS (Huo et al., 2021) algorithm, the coefficient $\eta$ between $10^{-2}$ and $10^{-4}$ is used, where smaller $\eta$ is used for larger batch sizes. We performed a hyperparameter search with an order of 10. Therefore careful hyperparameter tuning like (Nado et al., 2021) may improve the performance. On the other hand, $\eta = 10^{-2}$ is used for LALC in all cases, which yields satisfactory results. A learning rate of 0.1 is used for a batch size of 128 and it is linearly scaled with batch size (Goyal et al., 2017a). Please see Appendix B for further details.

LAMB (You et al., 2019), which is a (modified) LARS based on the Adam optimizer (Kingma & Ba, 2014) is not considered in these experiments. The Adam optimizer requires an additional hyperparameter have different characteristics, complicating its fair comparison with an SGD optimizer. For example, it is known to exhibit very fast convergence, but a basic SGD optimizer with momentum often outperforms it with a longer training schedule. Although the whole experiment can be repeated with an Adam-family optimizer, we do not deem it to be necessary.

The experimental results are depicted in Figure 4 and 5. LALC outperforms learning rate scaling methods in large batch training, and exhibits similar performance compared to WarmUp (Goyal et al., 2017b) in small batch training. Application of WarmUp to the LALC optimizer does not

---

**Algorithm 1** LARS (You et al., 2017)

---

**Require:** $w^l$ : Weight matrix of layer l
**Require:** $\gamma_t$ : Learning rate at step t
**Require:** $\eta$ : LARS coefficient
**Require:** $\beta$ : Weight decay rate
**Require:** $f$ : Gradient optimization algorithm (SGD, Adam, etc.)
**Require:** $\epsilon$ : Small constant to avoid a numerical error
  **while** $t < T$ for each layer l **do**
    $g_t^l \leftarrow \frac{dL}{dw_t^l}$
    $h_t^l \leftarrow f(g^l, w_t^l, \beta)$
    $\lambda_t^l \leftarrow \frac{\eta||w_t^l||}{||g_t^l||+\beta||w_t^l||+\epsilon}$
    $w_{t+1}^l \leftarrow w_t^l - \gamma_t \lambda_t^l h_t^l$
  **end while**

---

---

**Algorithm 2** LARS/LAMB (You et al., 2019)

---

**Require:** $\phi$ : A scaling function
    **while** $t < T$ for each layer l **do**
        $g_t^l \leftarrow \frac{dL}{dw_t^l}$
        $h_t^l \leftarrow f(g^l, w_t^l, \beta)$
        $\lambda_t^l \leftarrow \frac{\phi(||w_t^l||)}{||h_t^l||+\epsilon}$
        $w_{t+1}^l \leftarrow w_t^l - \gamma_t \lambda_t^l h_t^l$
    **end while**

---

**Algorithm 3** LALC (ours)

---

    **while** $t < T$ for each layer l **do**
        $g_t^l \leftarrow \frac{dL}{dw_t^l}$
        $h_t^l \leftarrow f(g^l, w_t^l, \beta)$
        $\lambda_t^l \leftarrow \frac{\eta||w_t^l||}{||h_t^l||+\epsilon}$
        $w_{t+1}^l \leftarrow w_t^l - \boldsymbol{min(\gamma_t, \lambda_t^l)}h_t^l$
    **end while**

---

make a significant difference. This is attributed to the fact that the primary function of WarmUp is the reduction of training instability during early stages of training, which is already handled by LALC. To the best of our knowledge, there is no general optimization theory that suggests using a much smaller learning rate at the beginning of training compared to that in the middle of training.

Nesterov momentum (Botev et al., 2016) improves the performance slightly but stably in a variety of setups. However, we could not find evidence that it particularly overcomes training instability. (Lin et al., 2020) proposed a method to improve Nesterov momentum—but it is targeted for distributed learning and no improvement is observed in the single-node case. Huo et al. (2021) is also targeted for distributed learning, but it also suggests using a per-sample gradient instead of a minibatch gradient. Some benefits are observed in certain cases, but the computational cost of calculating the norm of the per-sample gradient is not negligible.

## 5 DISCUSSION & FUTURE WORKS

In this paper, we analyze gradient explosion that occurs in general neural network architecture with ReLU activation and batch normalization. However, further theoretical analysis is required. For example, Theorem 2 states that gradient vanishing *may* occur if a high correlation exists between the inputs. Experiments on the simple MLP model (Figure 1(a)) may suggest that the equality condition (identical variance before batch normalization layer) is not very tight—but an upper bound is required to guarantee the gradient decrease theoretically. The problem is that there is a probability of divergence when the variance approaches zero. Thus, a small constant should be added to the estimated variance to avoid a numerical error (denoted by '$\epsilon$' in (Ioffe & Szegedy, 2015)). However, this makes the equation mathematically tricky—we intend to explore it in a future work. The convolutional layer is observed to exhibit a smaller explosion rate compared to fully connected layers (Figure 1(a)). This may be attributed to the correlation between inputs in the CNN since spatially adjusted pixels tend to exhibit similar activation.

A local learning rate of LARS (You et al., 2017) is used to select an upper bound of the learning rate. This is an engineering solution, which is not theoretically grounded. Although exact calculation is practically unrealistic, a practical approximation of the second derivative or the noise induced by the update may be estimated to determine a more appropriate upper bound.

Networks using layer normalization, including transformer-based models (Dosovitskiy et al., 2020; Liu et al., 2021), do not exhibit gradient explosion. This may partially contribute to the success of large-scale transformer models in various domains. However, it does not necessarily imply that batch normalization should be replaced in models of every type, since layer normalization may suffer from greater sampling noise depending on the network architecture.

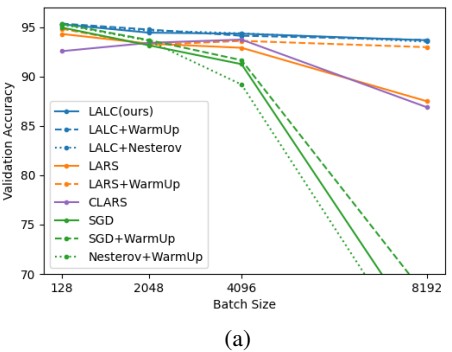 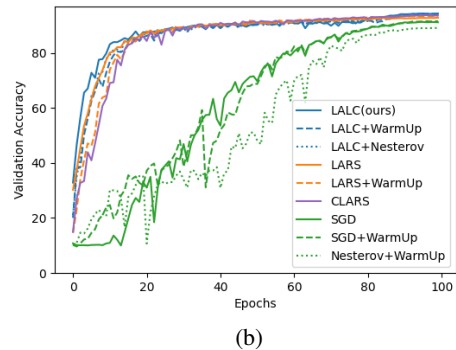

|         |         |
|---------|---------|
| (a)     | (b)     |

Figure 4: CIFAR10 experiment for (a) different batch sizes and (b) training curves for 4k batch size. LALC outperforms existing methods in large batch training, and exhibits similar performance compared to WarmUp (Goyal et al., 2017b) in small batch training. Application of WarmUp to LALC does not have a significant effect. We assume that one of the primary functions of WarmUp is to relieve the training instability, which is already handled by LALC. We could not observe any evidence that Nesterov momentum overcomes training instability. LARS and CLARS are sufficiently effective to handle the training instability, but they are applied throughout training at every layer and degrade performance slightly. Using excessively large batch size in the CIFAR10 experiment degrades performance irrespective of the algorithm since CIFAR10 only contain 50k training data points and the parameter can only be updated a number of times per epoch with batch sizes of 8k or 16k.

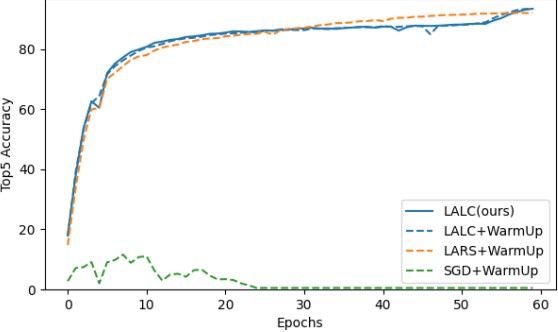

Figure 5: ImageNet experiment with 8k batch size. We obtained similar results and training curves to the CIFAR experiment. Both LALC and LARS effectively handled training instability at the early stage of training, but we also found a minor performance degradation in LARS. We also could not observe a significant difference when WarmUp is applied to LALC.

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

## A  APPENDIX

### A.1  PROOF OF PROPOSITION 1

We use the moment generating function and its properties.

$$
\begin{aligned}
M_X(t) :&= E[e^{tX}] \\
&= e^0 P(X=0) + \int_0^\infty e^{tx} \frac{1}{\sqrt{2\pi}\sigma} e^{-\frac{(x-\mu)^2}{2\sigma^2}} dx \\
&= P(N(\mu,\sigma^2) \le 0) + \int_0^\infty \frac{1}{\sqrt{2\pi}\sigma} e^{-\frac{(x-\mu-\sigma^2 t)^2 - 2\mu\sigma^2 t - \sigma^4 t^2}{2\sigma^2}} dx \\
&= \phi(-\frac{\mu}{\sigma}) + e^{\mu t + \frac{\sigma^2 t^2}{2}} \int_0^\infty \frac{1}{\sqrt{2\pi}\sigma} e^{-\frac{(x-\mu-\sigma^2 t)^2}{2\sigma^2}} dx
\end{aligned}
\tag{7}
$$

where $\phi(x) = \int_{-\infty}^x \frac{1}{\sqrt{2\pi}} e^{-x^2/2}$ is the cumulative distribution function of the standard normal distribution. Substituting $x' = (x - \mu - \sigma^2 t)/\sigma$,

$$
\begin{aligned}
&= \phi(-\frac{\mu}{\sigma}) + e^{\mu t + \frac{\sigma^2 t^2}{2}} \int_{-\frac{\mu+\sigma^2 t}{\sigma}}^\infty \frac{1}{\sqrt{2\pi}} e^{-\frac{x'^2}{2}} dx' \\
&= \phi(-\frac{\mu}{\sigma}) + e^{\mu t + \frac{\sigma^2 t^2}{2}} (1 - \phi(-\frac{\mu + \sigma^2 t}{\sigma}))
\end{aligned}
\tag{8}
$$

We calculate the first and second moment from the moment generating function.

$$
\begin{aligned}
\frac{d}{dt} M_X(t) &= (\mu + \sigma^2 t) e^{\mu t + \frac{\sigma^2 t^2}{2}} (1 - \phi(-\frac{\mu + \sigma^2 t}{\sigma})) + e^{\mu t + \frac{\sigma^2 t^2}{2}} (-1)(-\sigma) \frac{1}{\sqrt{2\pi}} e^{-\frac{1}{2}(\frac{\mu+\sigma^2 t}{\sigma})^2} \\
&= (\mu + \sigma^2 t) e^{\mu t + \frac{\sigma^2 t^2}{2}} (1 - \phi(-\frac{\mu + \sigma^2 t}{\sigma})) + \frac{\sigma}{\sqrt{2\pi}} e^{-\frac{\mu^2 + 2\mu\sigma^2 t + \sigma^4 t^2 - 2\sigma^2(\mu t + \sigma^2 t^2/2)}{2\sigma^2}} \\
&= (\mu + \sigma^2 t) e^{\mu t + \frac{\sigma^2 t^2}{2}} (1 - \phi(-\frac{\mu + \sigma^2 t}{\sigma})) + \frac{\sigma}{\sqrt{2\pi}} e^{-\frac{\mu^2}{2\sigma^2}}
\end{aligned}
\tag{9}
$$

$$
\begin{aligned}
\frac{d^2}{dt^2} M_X(t) &= (\sigma^2 + (\mu + \sigma^2 t)^2) e^{\mu + \frac{\sigma^2 t^2}{2}} (1 - \phi(-\frac{\mu + \sigma^2 t}{\sigma})) \\
&\quad + (\mu + \sigma^2 t)\, e^{\mu t + \frac{\sigma^2 t^2}{2}} \frac{d}{dt}[1 - \phi(-\frac{\mu + \sigma^2 t}{\sigma})] \\
&= (\sigma^2 + (\mu + \sigma^2 t)^2) e^{\mu t + \frac{\sigma^2 t^2}{2}} (1 - \phi(-\frac{\mu + \sigma^2 t}{\sigma})) + (\mu + \sigma^2 t) \frac{\sigma}{\sqrt{2\pi}} e^{-\frac{\mu^2}{2\sigma^2}}
\end{aligned}
\tag{10}
$$

Now, we finally obtain:

$$
E(X) = \frac{d}{dt} M_X(t)|_{t=0} = \mu(1 - \phi(-\frac{\mu}{\sigma})) + \frac{\sigma}{\sqrt{2\pi}} e^{-\frac{\mu^2}{2\sigma^2}}
\tag{11}
$$

$$
\begin{aligned}
Var(X) &= \frac{d^2}{dt^2} M_X(t)|_{t=0} - (\frac{d}{dt} M_X(t)|_{t=0})^2 \\
&= (\sigma^2 + \mu^2)(1 - \phi(-\frac{\mu}{\sigma})) + \mu \frac{\sigma}{\sqrt{2\pi}} e^{-\frac{\mu^2}{2\sigma^2}} - (\mu(1 - \phi(-\frac{\mu}{\sigma})) + \frac{\sigma}{\sqrt{2\pi}} e^{-\frac{\mu^2}{2\sigma^2}})^2
\end{aligned}
\tag{12}
$$

which proves the proposition.

## A.2 PROOF OF THEOREM 1

Let us denote $ReLU(x^n) = x^{n+}$, $W^n x^{n+} + b^n = \hat{x}^{n+1}$, $BatchNorm(\hat{x}^{n+1}) = x^{n+1}$.

First, we need to estimate the distribution of $\hat{x}^{n+1}$, which is estimated by the batch normalization layer.

$$
\begin{aligned}
\hat{\mu}_j^{n+1} &= E[\hat{x}_j^{n+1}] \\
&= E[\sum_i (x_i^{n+} W_{ij}^n) + b_j^n] \\
&= \sum_i (E[x_i^{n+}] W_{ij}^n) + b_j^n \\
&= \sum_i (\frac{\sigma_i^n}{\sqrt{2\pi}} W_{ij}^n) + b_j^n
\end{aligned}
\tag{13}
$$

This equation describes the sampling process of the batch normalization layer—the statistics of model parameters($W_{ij}^n$, $b_j^n$) are not considered. Thus, '$E$' denotes the expectation related to the input distribution. Similarly,

$$
\begin{aligned}
(\hat{\sigma}_j^{n+1})^2 &= Var(\hat{x}_j^{n+1}) \\
&= Var(\sum_i (x_i^{n+} W_{ij}^n) + b_j^n) \\
&= \sum_i Var(x_i^{n+} W_{ij}^n) \\
&= \sum_i (W_{ij}^n)^2 Var(x_i^{n+}) \\
&= \sum_i (W_{ij}^n)^2 ((\sigma_i^n)^2 (1 - \phi(0)) - \frac{(\sigma_i^n)^2}{2\pi} e^0) \\
&= (\frac{1}{2} - \frac{1}{2\pi}) \sum_i (\sigma_i^n)^2 (W_{ij}^n)^2
\end{aligned}
\tag{14}
$$

Recall that $x_j^{n+1}$ is given by:

$$
\begin{aligned}
x_j^{n+1} &= BatchNorm(\hat{x}^{n+1}) \\
&= \frac{\hat{x}^{n+1} - \hat{\mu}_j^{n+1}}{\hat{\sigma}_j^{n+1}} \gamma_j^{n+1} + \beta_j^{n+1}
\end{aligned}
\tag{15}
$$

where $\gamma^{n+1}, \beta^{n+1} \in \mathbb{R}^{n_{d+1}}$ are affine transform parameters of batch normalization.

Now, we calculate the gradient of the input using the backpropagation algorithm. Since ReLU operation not only changes the activation but also blocks the gradient, they are treated separately. For indices such that $x_i^n \leq 0$, the gradient $g_i^n$ is just zero. Then, for each $i$ that is not blocked by ReLU (i.e. for $i \in \{i : x_i^n > 0\}$), we have:

$$
\begin{aligned}
g_i^n = \frac{dL}{dx_i^n} &= \sum_j \frac{dL}{dx_j^{n+1}} \frac{dx_j^{n+1}}{dx_i^n} \\
&= \sum_j g_j^{n+1} \frac{dx_j^{n+1}}{d\hat{x}_j^{n+1}} \frac{d\hat{x}_j^{n+1}}{dx_i^n} \\
&= \sum_j g_j^{n+1} \frac{\gamma_j^{n+1}}{\hat{\sigma}_j^{n+1}} W_{ij}^n
\end{aligned}
\tag{16}
$$

$$
\begin{aligned}
Var(g_i^n) &= Var(\sum_j g_j^{n+1} \frac{\gamma_j^{n+1}}{\hat{\sigma}_j^{n+1}} W_{ij}^n) \\
&= \sum_j Var(g_j^{n+1} \frac{\gamma_j^{n+1}}{\hat{\sigma}_j^{n+1}} W_{ij}^n) \\
&= \sum_j Var(g_j^{n+1})(E[\frac{\gamma_j^2}{(\hat{\sigma}_j^{n+1})^2}(W_{ij}^n)^2] - E[\frac{\gamma_j^{n+1}}{\sigma_j^{n+1}} W_{ij}^n]^2) \\
&= \sum_j Var(g_j^{n+1})(\gamma_j^{n+1})^2 E[\frac{1}{(\hat{\sigma}_j^{n+1})^2}] E[(W_{ij}^n)^2]
\end{aligned}
\tag{17}
$$

Since $f(x) = 1/x$ is convex at $x > 0$, we can apply Jensen's inequality (Jensen, 1906).

$$
\begin{aligned}
&\geq \sum_j Var(g_j^{n+1})(\gamma_j^{n+1})^2 \frac{1}{E[(\hat{\sigma}_j^{n+1})^2]} E[(W_{ij}^n)^2] \\
&= \sum_j Var(g_j^{n+1})(\sigma_j^{n+1})^2 \frac{E[(W_{ij}^n)^2]}{E[(\frac{1}{2} - \frac{1}{2\pi}) \sum_k (\sigma_k^n)^2 (W_{kj}^n)^2]} \\
&= \frac{2\pi}{\pi - 1} \sum_j (\sigma_j^{n+1})^2 Var(g_j^{n+1}) \frac{\sigma_w^2}{E[\sum_k (\sigma_k^n)^2]\sigma_w^2} \\
&= \frac{2\pi}{\pi - 1} \frac{1}{\sum_k (\sigma_k^n)^2} \sum_j (\sigma_j^{n+1})^2 Var(g_j^{n+1})
\end{aligned}
\tag{18}
$$

The equality holds when $\hat{\sigma}_j^{n+1}$ is identical for all $j$. Note that this formula is no longer dependent on the input index $i$ if $x_i^n$ is not blocked by ReLU. Moreover, as the input distribution is zero-centered by assumption, the probability of passing ReLU is $1/2$ for all $i$, independent of $\sigma_i^n$. Therefore, we can arrange the equation:

$$
\begin{aligned}
&E[\sum_i (\sigma_i^n)^2 Var(g_i^n)] / \sum_j (\sigma_j^{n+1})^2 Var(g_j^{n+1} \\
&\geq \frac{1}{2} E[\sum_i (\sigma_i^n)^2 \frac{2\pi}{\pi - 1} \frac{1}{\sum_k (\sigma_k^n)^2} \sum_j (\sigma_j^{n+1})^2 Var(g_j^{n+1}) / \sum_j (\sigma_j^{n+1})^2 Var(g_j^{n+1}] \\
&= \frac{1}{2} \frac{2\pi}{\pi - 1} \frac{\sum_i (\sigma_i^n)^2}{\sum_k (\sigma_k^n)^2} \\
&= \frac{\pi}{\pi - 1}
\end{aligned}
\tag{19}
$$

This proves Theorem 1.

### A.3   PROOF OF COROLLARY 2

Using the same notations as Theorem 1, let $I^+ := \{i | \mu_i > 0\}$. We have:

$$\lim_{\sigma_i^n/\mu_i^n \to 0 \, \forall i} (\hat{\sigma}_j^{n+1})^2 = \lim_{\sigma_i^n/\mu_i^n \to 0 \, \forall i} Var(\hat{x}_j^{n+1})$$

$$= \lim_{\sigma_i^n/\mu_i^n \to 0 \, \forall i} \sum_i (W_{ij}^n)^2 Var(x_i^{n+})$$

$$= \lim_{\sigma_i^n/\mu_i^n \to 0 \, \forall i} \sum_i (W_{ij}^n)^2 (((\sigma_i^n)^2 + (\mu_i^n)^2)(1 - \phi(-\frac{\mu_i^n}{\sigma_i^n})) + \mu_i^n \frac{\sigma_i^n}{\sqrt{2\pi}} e^{-\frac{1}{2}(\frac{\mu_i^n}{\sigma_i^n})^2}$$

$$- (\mu_i^n(1 - \phi(-\frac{\mu_i^n}{\sigma_i^n})) + \frac{\sigma_i^n}{\sqrt{2\pi}} e^{-\frac{1}{2}(\frac{\mu_i^n}{\sigma_i^n})^2})^2)$$

$$= \sum_{i \in I^+} (W_{ij}^n)^2 (((\sigma_i^n)^2 + (\mu_i^n)^2)(1 - 0) + \mu_i^n \frac{\sigma_i^n}{\sqrt{2\pi}} \cdot 0 - (\mu_i^n(1 - 0) + \mu_i^n \frac{\sigma_i^n}{\sqrt{2\pi}} \cdot 0)^2)$$

$$+ \sum_{i \notin I^+} (W_{ij}^n)^2 (((\sigma_i^n)^2 + (\mu_i^n)^2)(1 - 1) + \mu_i^n \frac{\sigma_i^n}{\sqrt{2\pi}} \cdot 0 - (\mu_i^n(1 - 1) + \mu_i^n \frac{\sigma_i^n}{\sqrt{2\pi}} \cdot 0)^2)$$

$$= \sum_{i \in I^+} (W_{ij}^n)^2 ((\sigma_i^n)^2 + (\mu_i^n)^2 - (\mu_i^n)^2) + \sum_{i \notin I^+} (W_{ij}^n)^2 \cdot 0$$

$$= \sum_{i \in I^+} (\sigma_i^n)^2 (W_{ij}^n)^2$$

$$(20)$$

The remainder of the proof is similar to that of Theorem 1.

## A.4 PROOF OF THEOREM 2

Since inputs($x^n$) are perfectly correlated, let $b \sim N(0, 1)$ and $x^n = bt$, where $t \in \mathbb{R}^{n_d}$ is treated as a constant. Let $ReLU(x^n) = x^{n+}$, $W^n x^{n+} + b^n = \hat{x}^{n+1}$, and $BatchNorm(\hat{x}^{n+1}) = x^{n+1}$. We first calculate the mean and variance of $\hat{x}^{n+1}$, which are estimated by the batch normalization layer. As the sign of $x^n$ is reversed corresponding to positive and negative $t$, they are treated separately.

$$\hat{\mu}_j^{n+1} = E[\hat{x}_j^{n+1}]$$

$$= E[\hat{x}_j^{n+1}|b \geq 0]P(b \geq 0) + E[\hat{x}_j^{n+1}|b \leq 0]P(b \leq 0)$$

$$= \frac{1}{2} E[\sum_i W_{ij}^n ReLU(t_i) b | b \geq 0] + \frac{1}{2} E[\sum_i W_{ij}^n ReLU(-t_i)(-b) | b \leq 0]$$

$$= \frac{1}{2} \frac{2}{\sqrt{2\pi}} \sum_i W_{ij}^n (ReLU(t_i) + ReLU(-t_i))$$

$$= \frac{1}{\sqrt{2\pi}} \sum_i W_{ij}^n |t_i|$$

$$(21)$$

The conditional expectation of $b$ is calculated using Proposition 1. Then, we directly calculate the variance before normalization.

$$(\hat{\sigma}_j^{n+1})^2 = \int_{-\infty}^{\infty} (\hat{x}_j^{n+1} - \hat{\mu}_j^{n+1})^2 P(b) db$$

$$= \int_{-\infty}^{0} (\sum_i W_{ij}^n ReLU(-t_i)(-b) - \hat{\mu}_j^{n+1})^2 \frac{1}{\sqrt{2\pi}} e^{-\frac{b^2}{2}} db$$

$$+ \int_{0}^{\infty} (\sum_i W_{ij}^n ReLU(t_i)(b) - \hat{\mu}_j^{n+1})^2 \frac{1}{\sqrt{2\pi}} e^{-\frac{b^2}{2}} db$$

$$= \int_{-\infty}^{0} (-N_j b - \hat{\mu}_j^{n+1})^2 \frac{1}{\sqrt{2\pi}} e^{-\frac{b^2}{2}} db + \int_{0}^{\infty} (P_j b - \hat{\mu}_j^{n+1})^2 \frac{1}{\sqrt{2\pi}} e^{-\frac{b^2}{2}} db$$

$$(22)$$

where $N_j := \sum_i W_{ij}^n ReLU(-t_i)$ and $P_j := \sum_i W_{ij}^n ReLU(t_i)$. Using the formula $\int (ax - b)^2 e^{-x^2/2} dx = \sqrt{\frac{\pi}{2}}(a^2 + b^2)erf(\frac{x}{\sqrt{2}}) - ae^{-x^2/2}(ax - 2b) + C$, where $erf(x) = 2\phi(x\sqrt{2}) - 1$ is an error function, we obtain:

$$
\begin{aligned}
&= \frac{1}{\sqrt{2\pi}}\Big[\sqrt{\frac{\pi}{2}}(N_j^2 + (\hat{\mu}_j^{n+1})^2)erf(\frac{x}{\sqrt{2}}) + Ne^{-\frac{x^2}{2}}(-N_j x - 2\hat{\mu}_j^{n+1})\Big]_{-\infty}^0 \\
&\quad + \frac{1}{\sqrt{2\pi}}\Big[\sqrt{\frac{\pi}{2}}(P_j^2 + (\hat{\mu}_j^{n+1})^2)erf(\frac{x}{\sqrt{2}}) - Pe^{-\frac{x^2}{2}}(P_j x - 2\hat{\mu}_j^{n+1})\Big]_0^\infty \\
&= \frac{1}{\sqrt{2\pi}}[-2N_j\hat{\mu}_j^{n+1} + \sqrt{\frac{\pi}{2}}(N_j^2 + (\hat{\mu}_j^{n+1})^2) + \sqrt{\frac{\pi}{2}}(P_j^2 + (\hat{\mu}_j^{n+1})^2) - 2P_j\hat{\mu}_j^{n+1}] \\
&= \frac{1}{2}(N_j^2 + P_j^2) + (\hat{\mu}_j^{n+1})^2 - \sqrt{\frac{2}{\pi}}\hat{\mu}_j^{n+1}(P_j + N_j)
\end{aligned}
\tag{23}
$$

Since $\{W_{ij}\}$ are independent of each other and $E[N_j] = E[P_j] = E[\hat{\mu}_j^{n+1}] = 0$, we obtain:

$$
\begin{aligned}
E[(\hat{\sigma}_j^{n+1})^2] &= E[\frac{1}{2}(N_j^2 + P_j^2) + (\mu_j^{n+1})^2 - \sqrt{\frac{2}{\pi}}\mu_j^{n+1}(P_j + N_j)] \\
&= \frac{1}{2}(E[N_j^2] + E[P_j^2]) + E[(\mu_j^{n+1})^2] - \sqrt{\frac{2}{\pi}}E[\mu_j^{n+1}(P_j + N_j)] \\
&= \frac{1}{2}(Var(N_j) + Var(P_j)) + Var(\mu_j^{n+1}) - 0 \\
&= \frac{1}{2}(\sigma_w^2 \sum_i ReLU(t_i)^2 + \sigma_w^2 \sum_i ReLU(-t_i)^2) + \frac{1}{2\pi}\sigma_w^2 \sum_i |t_i|^2 \\
&= \sigma_w^2(\frac{1}{2} + \frac{1}{2\pi})\sum_i t_i^2
\end{aligned}
\tag{24}
$$

where $\sigma_w$ denotes the standard deviation of the weights. Let $I^+ := \{i : x_i^n > 0\}$. For $i \in I^+$, we have:

$$
\begin{aligned}
Var(g_i^n) &= Var(\sum_j g_j^{n+1}\frac{\gamma_j^{n+1}}{\hat{\sigma}_j^{n+1}}W_{ij}^n) \\
&= \sum_j Var(g_j^{n+1}\frac{\gamma_j^{n+1}}{\hat{\sigma}_j^{n+1}}W_{ij}^n) \\
&= \sum_j Var(g_j^{n+1})(E[\frac{\gamma_j^2}{(\hat{\sigma}_j^{n+1})^2}(W_{ij}^n)^2] - E[\frac{\gamma_j^{n+1}}{\sigma_j^{n+1}}W_{ij}^n]^2) \\
&= \sum_j Var(g_j^{n+1})(\gamma_j^{n+1})^2 E[\frac{1}{(\hat{\sigma}_j^{n+1})^2}]E[(W_{ij}^n)^2] \\
&\geq \sum_j Var(g_j^{n+1})(\gamma_j^{n+1})^2 \frac{1}{E[(\hat{\sigma}_j^{n+1})^2]}E[(W_{ij}^n)^2] \\
&= \frac{2\pi}{\pi + 2}\sum_j (\sigma_j^{n+1})^2 Var(g_j^{n+1})\frac{\sigma_w^2}{\sum_k t_k^2 \sigma_w^2} \\
&= \frac{2\pi}{\pi + 2}\frac{1}{\sum_k t_k^2}\sum_j (\sigma_j^{n+1})^2 Var(g_j^{n+1})
\end{aligned}
\tag{25}
$$

The equality holds when $\hat{\sigma}_j^{n+1}$ are identical for all $j$. Finally, we obtain:

$$E[\sum_i (\sigma_i^n)^2 Var(g_i^n) / \sum_j (\sigma_j^{n+1})^2 Var(g_j^{n+1})]$$

$$\geq E[\frac{1}{2} \sum_i t_i^2 \frac{2\pi}{\pi+2} \frac{1}{\sum_k t_k^2} \sum_j (\sigma_j^{n+1})^2 Var(g_j^{n+1}) / \sum_j (\sigma_j^{n+1})^2 Var(g_j^{n+1})] \tag{26}$$

$$= \frac{1}{2} \frac{2\pi}{\pi+2} \frac{\sum_i t_i^2}{\sum_k t_k^2}$$

$$= \frac{\pi}{\pi+2}$$

This proves the theorem.

## B    EXPERIMENTAL DETAILS

SGD with a momentum of 0.9 and weight decay with a rate of $5 \cdot 10^{-4}$ are used in all experiments. In the CIFAR10 experiment, ResNet50 (He et al., 2016) architecture is used, in which the first pooling layer and stride operation of the first convolution layer are removed to handle a smaller image size. The input is normalized using the mean = (0.4914, 0.4822, 0.4465) and std = (0.2023, 0.1994, 0.2010). Random cropping with a padding size of 4 is used and random horizontal flips are applied. When WarmUp (Goyal et al., 2017b) is used, the learning rate is linearly increased over the first 10 (CIFAR10) or two epochs (ImageNet). After that, cosine learning rate decay (Loshchilov & Hutter, 2016) without restarts is used.

In the ImageNet experiment, input is normalized using the mean = (0.485, 0.456, 0.406) and std = (0.229, 0.224, 0.225). RandomResizedCrop (He et al., 2016) with an image size of 224 is used as augmentation. Then, a random horizontal flip is applied. In the evaluation step, images are resized to the size 256 and center-cropped to size 224.

In experiments with very large batch sizes, a computational trick is used to updates the parameter every n steps to increase the batch size by n times effectively. It is almost identical to using a batch size that is n times larger, except that it involves slightly higher sampling errors from normalization layers.

The CLARS algorithm requires a per-sample gradient, which cannot be easily obtained in popular deep learning frameworks, such as PyTorch (Paszke et al., 2019) or Tensorflow (Abadi et al., 2015). Thus, a module is borrowed to obtain the per-sample gradient from Opacus (Yousefpour et al., 2021), which was originally developed for differential privacy. This is the best option to the best of our knowledge, but it requires twice the time and memory.

| Batch size | LARS | CLARS | LALC(ours) |
|------------|------|-------|------------|
| 128 | 0.01 | 0.01 | 0.01 |
| 2048 | 0.001 | 0.001 | 0.01 |
| 4096 | 0.001 | 0.001 | 0.01 |
| 8192 | 0.001 | 0.0001 | 0.01 |

Table 1: Coefficient $\eta$ used in our experiment. A hyperparameter search in the order of 10 is performed.

```python
for p in group['params']:
    # Optimizer part. It can be any optimizer you want.
    if p.grad is None:
        continue
    d_p = p.grad.data
    if weight_decay != 0:
        d_p.add_(weight_decay, p.data)
    if momentum != 0:
        param_state = self.state[p]
        if 'momentum_buffer' not in param_state:
            buf = param_state['momentum_buffer'] = torch.clone(d_p).detach()
        else:
            buf = param_state['momentum_buffer']
        buf.mul_(momentum).add_(1 - dampening, d_p)
        if nesterov:
            d_p = d_p.add(momentum, buf)
        else:
            d_p = buf

    # LALC part
    w_norm = torch.norm(p.data) # L2 norm in default
    g_norm = torch.norm(d_p)
    if w_norm * g_norm > 0:
        local_lr = eta * w_norm / (g_norm+epsilon)
        local_lr = local_lr.cpu().detach().item()
        # Becomes very similar to LARS if this part is "lr=local_lr*group['lr']"
        lr = min(local_lr, group['lr'])
    else:
        lr = group['lr']

    # Update the parameter
    p.data.add_(-lr, d_p)
```

Figure 6: PyTorch (Paszke et al., 2019) implementation of LALC. It can easily be implemented by adding a few lines of code to the existing optimizer.

| Method | Batch size | | | |
|---|---|---|---|---|
| | 128 | 2048 | 4096 | 8192 |
| SGD | 94.97 | 93.17 | 91.26 | 63.12 |
| SGD+WarmUp | 95.35 | 93.7 | 91.66 | 67.67 |
| Nesterov+WarmUp | 95.19 | 93.63 | 89.19 | 60.24 |
| LARS | 94.32 | 93.34 | 92.92 | 87.5 |
| LARS+WarmUp | 94.84 | 93.21 | 93.62 | 92.98 |
| CLARS | 92.58 | 93.42 | 93.75 | 86.89 |
| LALC(ours) | 95.3 | 94.45 | 94.35 | 93.68 |
| LALC+WarmUp | 95.37 | 94.77 | 94.14 | 93.71 |
| LALC+Nesterov | 95.22 | 94.67 | 94.39 | 93.71 |

Table 2: The validation accuracy in the CIFAR10 experiment (Figure 4).

