# OpenReview forum: "Training Instability and Disharmony Between ReLU and Batch Normalization"
_ICLR.cc/2023/Conference — Submitted to ICLR 2023_

### Official Review · Reviewer_NA9i · 2022-10-22

**Confidence:** 3
**Correctness:** 3
**Technical Novelty And Significance:** 2
**Empirical Novelty And Significance:** 2
**Recommendation:** 3

**Clarity, Quality, Novelty And Reproducibility:**

Reproducibility, writing clarity and quality is good. There are some questions on novelty (see above).


**Strength And Weaknesses:**

Strengths (S):

----------------

S: The studied problem of exploding gradients problem is a very important one in deep learning, expecially since it happens with the most typical combination of normalization and activation functions, namely ReLU and BN.

S: The paper proposes a simple extension (LALC) to LARS optimizer that seems to fix at least partially the exploding gradients problem by clipping the learning rate.

S: If LALC is shown to be better than LAMB and the key reference comment (see below) is correct, the paper could be revised to focus on LALC.

Weaknesses (W):

----------------

W: The main weakness of the paper seems to be missing key references. Could the authors discuss how the analytical part of the work is different from this paper Yang, Greg, et al. "A mean field theory of batch normalization." arXiv preprint arXiv:1902.08129 (2019). https://arxiv.org/abs/1902.08129 and this posting: https://kyleluther.github.io/2020/02/18/batchnorm-exploding-gradients.html.

W: Regarding the clipping method (LALC), the paper seems to fail to cite Fong, Jeffrey, Siwei Chen, and Kaiqi Chen. "Improving Layer-wise Adaptive Rate Methods using Trust Ratio Clipping." arXiv preprint arXiv:2011.13584 (2020). https://arxiv.org/abs/2011.13584 , which also contains clipping. Please compare both analytically and experimentally to this method.

W: The experimental part should contain trainings with LAMB, since in many cases it improves over LARS without expensive tuning of LAMB hyperparams.

W: The results with LALC could be compared with the 3 proposed solutions in https://arxiv.org/abs/1902.08129



**Summary Of The Paper:**

This paper studies the incompatibility of two very commonly found components of deep learning, namely the ReLU and Batch Normalization. This incompatibility is manifested e.g., in exploding gradients during the early part of training. The authors hypothesize that the source of the incompatibility is the fact that the variance of the activations and their gradients is affected differently by the combination of ReLU and Batch Normalization. The authors show this analytically, then explore it experimentally. Finally, they propose a solution to alleviate the problem.


**Summary Of The Review:**

Decent work but key references missing.

---

> ### Author Response · Authors · 2022-11-17
> **Responses to the Reviewer NA9i's Comments**
>
> We are grateful for your thorough consideration and review of our paper. We received a lot of valuable advice about the weakness of our paper and how to develop our research. Especially, we got multiple comments about the novelty and theoretical background of our solution. Since we better understood the cause of the problem, we should have proposed a better solution based on it. We are trying to extend our theoretical analysis. Although the gradient exponentially grows w.r.t. depth, The second-order derivative seems to diverge even faster. Maybe deriving the step size considering this would give us a better solution. Again, we are grateful for your insightful comments to improve our work.
>
> * Reviewer's comment: The main weakness of the paper seems to be missing key references. Could the authors discuss how the analytical part of the work is different from this paper Yang, Greg, et al. "A mean field theory of batch normalization." arXiv preprint arXiv:1902.08129 (2019). https://arxiv.org/abs/1902.08129 and this posting: https://kyleluther.github.io/2020/02/18/batchnorm-exploding-gradients.html.
>    * Author's response: Thank you for introducing us to an important reference we missed. The paper seemed to be a completely different approach to the same phenomenon, which has pros and cons compared to our approach. The paper covers arbitrary activation functions that have Gegenbauer expansion, so it would be more informative for those who want to improve the activation function to handle this problem. But since the paper is based on the mean-field theory, they could not handle the correlation between inputs, and therefore could not analyze how the gradient exploding problem is alleviated during training. Also, Their approach used more assumptions compared to ours, like infinite width and normally distributed weight parameters.
> * Reviewer's comment: The results with LALC could be compared with the 3 proposed solutions in https://arxiv.org/abs/1902.08129
>    * Author's response: Thank you for your suggestion. We agree with you, and we will include the experiment of those algorithms in the paper.

---

> > ### Comment · Reviewer_NA9i · 2022-12-12
> > **Thank you**
> >
> > Thank you for your rebuttal. I have decided to keep my score.

---

### Official Review · Reviewer_eGDm · 2022-10-22

**Confidence:** 3
**Correctness:** 3
**Technical Novelty And Significance:** 2
**Empirical Novelty And Significance:** 3
**Recommendation:** 3

**Clarity, Quality, Novelty And Reproducibility:**

Clarity:

In general I find this paper could improve a lot to tell a coherent story such that readers can more easily follow. Examples:

1. I cannot tell a clear connection of the theoretical results with the proposed algorithm: it is true that we need some layerwise changes based on the insights of the theory presented, but as also the authors admit, previous works like LARS/LAMB could have addressed that. Therefore it would be much better to state more clearly about, what is connection between the specific change proposed on top of LARS/LAMB (i.e., Algorithm 3) and the main theory in the previous sections.

2. I also find it difficult to understand what is the unique advantage of the proposed change over LARS/LAMB, even if we do not force its connected with the theory. For example section 3 is full of (a little bit) lengthy texts. Could the authors state it in a more structured manner (e.g., bullets, sub-sections with clear titles) to show the "logic flow"?

3. Some other obvious flaws like please use the correct format of left quotation mark in Latex, and "However, Contrary to popular belief" -> "contrary"


Novelty:

I'm indeed thinking the novelty of this paper mainly lying in the mathematical proof to reveal the gradient explosion order of BN + RELU under some restricted settings. However as aforementioned, I do not see a clear connection of this with the proposed solution so this does limit the real contrition of it.

Instead figure 1 (a) looks more interesting and I'm not sure whether it can act as the true novelty (despite needs more works though): is there a universal theoretical framework to tell people that under what circumstances (e.g., LN v.s. BN, self-attention v.s. CNN, etc) that the gradient explosion norm would grow in what ways? Note some are increasing while others are decreasing w.r.t. layers.  Furthermore more analysis to demonstrate the "adaptivity" of the previous algorithm like LARS/LAMB, for these possible theoretical analysis,  might be more interesting.



**Strength And Weaknesses:**

Strength:

1. This paper contains a certain level of insights: it theoretically analyzes the instability property of BN + RELU, and reveals that the order of speed of gradient explosion (defined by Var(X_l)Var(g_l)/Var(x_{l+1})Var(g_{l+1}). Its mathematical proof looks accurate and rigorous.

2. The proposed method is simple which is a good property for real-world applications, especially in industry level dataset/scenarios. Also given the simpleness, I'm in favor of its good reproducibility.

Weakness:

1. The written of this paper could be non-trivially improved, in particular the "logic flow" of it seems vague (detailed comments see below);

2. The mathematical conclusion of this paper (as its main technical contribution) seems narrow in that a) it limits to BN + RELU + no residual connection; b) it seems not closely related with the derived algorithm. Again check below for the details.

**Summary Of The Paper:**

This paper shows that mathematically the gradient explosion of Relu + BN network approximately grows in the speed of \sqrt{\pi/(\pi + 1)} w.r.t. to the layers. They also proposed a simple variant of LARS/LAMB that upperbounds the learning rate but not scale it. The authors conducted several experiments to show that the proposed method is effective.

**Summary Of The Review:**

Despite that this paper provides certain insights in terms of theoretical analysis of the "gradient explosion" pattern for RELU + BN, it is not stated clearly, and the novelty seems not being properly "positioned", in that 1) no clear reasoning leading to the empirical algorithm; 2) it's limited to a restricted scenario.

---

> ### Author Response · Authors · 2022-11-17
> **Responses to the Reviewer eGDm's Comments**
>
> We are grateful for your thorough consideration and review of our paper. We received a lot of valuable advice about the weakness of our paper and how to develop our research. Especially, we got multiple comments about the novelty and theoretical background of our solution. Since we better understood the cause of the problem, we should have proposed a better solution based on it. We are trying to extend our theoretical analysis. Although the gradient exponentially grows w.r.t. depth, The second-order derivative seems to diverge even faster. Maybe deriving the step size considering this would give us a better solution. Again, we are grateful for your insightful comments to improve our work.
>
> * Reviewer's comment:  is there a universal theoretical framework to tell people that under what circumstances (e.g., LN v.s. BN, self-attention v.s. CNN, etc) that the gradient explosion norm would grow in what ways? Note some are increasing while others are decreasing w.r.t. layers. Furthermore more analysis to demonstrate the "adaptivity" of the previous algorithm like LARS/LAMB, for these possible theoretical analysis, might be more interesting.
>    * Author's response: Thank you for your suggestion. Since a convolutional layer can be seen as a special case of a fully connected layer with sparse connectivity, we believe that the core part of the analysis will still be applied. We assume that the difference between CNN and transformer-based architecture comes from the difference in the normalization methods. But as you pointed out, we did not explain it enough. We will add a theoretical analysis that shows why the gradient does not explode under Layer Normalization. We will also try to analyze the effect and limitations of previous methods like LARS/LAMB.
>
> We are also grateful for the thoughtful comments on the writings of the paper.

---

### Official Review · Reviewer_EVtb · 2022-10-24

**Confidence:** 4
**Correctness:** 2
**Technical Novelty And Significance:** 2
**Empirical Novelty And Significance:** 2
**Recommendation:** 3

**Clarity, Quality, Novelty And Reproducibility:**

Paper is well written and their theoretical contribution is novel. The proposed algorithm and experimental settings are well explained.

**Strength And Weaknesses:**

**Strength**

The paper well-motivates the gradient explosion in presence of BN; then, it scientifically studies the cause of the explosion. The extensive references and linking of the results well to the literature are very well done.

**Weakness**

I have two main concerns about the theoretical and practical aspects of the results.
- *Theoretical.*
  - I am not sure the expectation in Thm.1 is taken w.r.t which random variables. What is the random variable in Eq. (3)?
  - I have not checked the proof but suppose that Thm.1. is soundly proven.  I am not sure Thm 1. obtains exponential growth. The  Thm. 1 states $E[a_n/a_{n+1}]>1$ where $a_n$ is a factor of gradient variance in layer $n$. Although it is stated such lower-bound in expectation leads to an exponential growth in $a_n$, I am not sure about this. If there was no expectation, it would be possible to derive exponential growth. However, the expectation makes the recurrence complicated. Would the authors mind if I ask to elaborate on this?
  - My experiments show the norm of the gradient in early layers (close to input) grows at an exponential rate with depth. While the paper suggests an exponential growth in the norm of the gradient in deep layers (close to outputs).
- *Practical.* The proposed algorithm is very similar to adaptive gradient clipping proposed by Brock, A., De, S., Smith, S. L., and Simonyan, K. in a paper titled "High-performance large-scale image
recognition without normalization".
What is the difference between the proposed method and the adaptive gradient clipping?


**Summary Of The Paper:**

 An important research question in neural computing is the interplay between Batch Normalization (BN) and learning. In this vein, this paper studies the influence of BN on gradient norm across the layers. BN causes an exponential growth with depth for the norm of a gradient in the first layer. When studying this phenomenon, the paper establishes theoretical and practical contributions:
- *Theory.* It is claimed that theorem 1. establishes an exponential growth in the norm of the gradient
- *Practice." Paper proposes a layer-wise stepsize clipping to compensate for the gradient expansion. The proposed method is effective in the classification of CIFAR10 and Imagenet datasets.

**Summary Of The Review:**

In short, I have some concerns regarding the soundness of theoretical contributions, and also some questions regarding the novelty of the proposed algorithm.

---

> ### Author Response · Authors · 2022-11-17
> **Responses to the Reviewer EVtb's Comments**
>
> We are grateful for your thorough consideration and review of our paper. We received a lot of valuable advice about the weakness of our paper and how to develop our research. Especially, we got multiple comments about the novelty and theoretical background of our solution. Since we better understood the cause of the problem, we should have proposed a better solution based on it. We are trying to extend our theoretical analysis. Although the gradient exponentially grows w.r.t. depth, The second-order derivative seems to diverge even faster. Maybe deriving the step size considering this would give us a better solution. Again, we are grateful for your insightful comments to improve our work.
>
> * Reviewer's comment: I am not sure the expectation in Thm.1 is taken w.r.t which random variables. What is the random variable in Eq. (3)?
>    * Author's response: It is an expectation about the distribution of input and weight initialization. We appreciate your comment, and we will clarify the meaning of the expectation throughout the paper.
> * Reviewer's comment: I have not checked the proof but suppose that Thm.1. is soundly proven. I am not sure Thm 1. obtains exponential growth. The Thm. 1 states E[an/an+1]>1 where an is a factor of gradient variance in layer n. Although it is stated such lower-bound in expectation leads to an exponential growth in an, I am not sure about this. If there was no expectation, it would be possible to derive exponential growth. However, the expectation makes the recurrence complicated. Would the authors mind if I ask to elaborate on this?
>    * Author's response:  As you pointed out, there exists a possibility that gradient explosion does not occur. For example, a ReLU network can be pseudo-linearized if every signal before the activation layer always has the same sign by chance. But this is statistically very unlikely for the neural network with thousands of dimensions. And we believe that our repeated experiments and observation would support this.
> * Reviewer's comment: My experiments show the norm of the gradient in early layers (close to input) grows at an exponential rate with depth. While the paper suggests an exponential growth in the norm of the gradient in deep layers (close to outputs).
>    * Author's response: You are right. Early layers have larger gradient norms under gradient explosion. We reversed the x-axis of figure 1 and 3, and this seems to confuse. We appreciate your comment, and we will clarify that part.

---

### Official Review · Reviewer_dvc8 · 2022-10-24

**Confidence:** 4
**Correctness:** 3
**Technical Novelty And Significance:** 3
**Empirical Novelty And Significance:** 3
**Recommendation:** 6

**Clarity, Quality, Novelty And Reproducibility:**

Clarity: The paper is well written, though there's a weak connection between theoretical part and solution part;

Novelty: The theoretical part is novel and interesting; The method is effective according to experimental results;

Reproducibility: One of the standard experimental result is not consistent with literature, as I point out in weakness.

**Strength And Weaknesses:**

Strength:

1) The theoretical result is interesting and novel, and match the empirical observations to some extents;

2) Experimental results prove the effectiveness of the proposed method;

Weakness:

1) As far as I concern, there is no apparent connection between the theoretical results (Sec 1-2) and proposed solution (Sec 3-4), which severely damage the integrity of the whole paper. I suggest the author to refer to a missing literature [1], in which another relationship between weights' norm and gradients' norm is derived. I think combining with the theoretical results in [1], the theoretical part can well support the reasonableness of the proposed method.

2) It is wired figure 5 shows SGD+warmup totally failed on imagenet with 8K batch size. In [2], the performance of Resnet50 on imagenet with 8K batch only degrades slightly. Can author explain why?

[1] Wan, Ruosi, et al. "Spherical Motion Dynamics: Learning Dynamics of Normalized Neural Network using SGD and Weight Decay." Neurips (2021): 6380-6391.

[2] Goyal, Priya, et al. "Accurate, large minibatch sgd: Training imagenet in 1 hour." arXiv preprint arXiv:1706.02677 (2017).



**Summary Of The Paper:**

This paper provides a new interpretations on why gradient explosion still can exist during the early training of normalized model. Their theoretical results can match the empirical observations to some extents. Based on their method, they propose a new optimization method, LALC, which can obtain good performance in both small batch and large batch training settings.

**Summary Of The Review:**

Overall speaking, the result is novel and interesting especially the theoretical part. But the theoretical part and solution part has weak connection. I suggest the author to compensate the gap by combining the theorems in the missing literature. Besides, re-examine the experiment setting.

---

> ### Author Response · Authors · 2022-11-17
> **Responses to the Reviewer dvc8's Comments**
>
> We are grateful for your thorough consideration and review of our paper. We received a lot of valuable advice about the weakness of our paper and how to develop our research. Especially, we got multiple comments about the novelty and theoretical background of our solution. Since we better understood the cause of the problem, we should have proposed a better solution based on it. We are trying to extend our theoretical analysis. Although the gradient exponentially grows w.r.t. depth, The second-order derivative seems to diverge even faster. Maybe deriving the step size considering this would give us a better solution. Again, we are grateful for your insightful comments to improve our work.
>
> * Reviewer's comment: It is wired figure 5 shows SGD+warmup totally failed on imagenet with 8K batch size. In [2], the performance of Resnet50 on imagenet with 8K batch only degrades slightly. Can author explain why?
>    * Author's response: You have raised an important question, which is not fully explained in the paper. Gradient explosion exists only at the early stage of the training. Even if we do not handle the problem directly, training with an infinitesimal learning rate with an extremely long training schedule would also overcome training instability. (Although it will require more resources as the instability increases.) We appreciate your comment, and we will add an experimental result that shows that a longer schedule of warmup further alleviates the problem but eventually reach the limit.

---

### Decision · Program_Chairs · 2023-01-20

**Decision:**

Reject

**Justification For Why Not Higher Score:**

The reviewers note that the connections between the theoretical results and the proposed solution is weak, and that the novelty of the paper is limited. In addition, some key references are missing.

**Justification For Why Not Lower Score:**

N/A

**Metareview: Summary, Strengths And Weaknesses:**

Ratings: 6/3/3/3
Confidences: 4/4/3/3
Recommendation: Reject

This paper provides a theoretical and practical solution to the incompatibility between ReLU and Batch Normalization, which is manifested in exploding gradients during the early part of training. The paper contains some interesting insights, and proposes the Layer-wise Asymmetric Learning rate Clipping (LALC) algorithm, an alternative to existing learning rate scaling methods in large batch training, and can also be used to replace WarmUp in small batch training. The reviewers note that the connections between the theoretical results and the proposed solution is weak, and that the novelty of the paper is limited. In addition, some key references are missing. The authors provided a rebuttal, but there was no further discussion.